# Tracing the Origin of Japan's First Buddhist Temple: Japan's Asukadera Viewed through the Lens of the Korean Paekche Kingdom Temple Site of Wanghŭng-sa

Byongho Lee

Department of Social Studies Education, Gongju National University of Education, Gongju 32553, Korea; gudara90@gmail.com

**Abstract:** The beginning of the construction of Japan's first large-scale Buddhist temple, Asukadera 飛鳥寺, in 588, marks an important turning point in the ancient history of Japan. The construction of a Buddhist temple was a major event through which Japanese people, who had believed in traditional indigenous religion, came to embrace Buddhism, one of the major world religions. Inscribed śarīra reliquaries were discovered from the wooden pagoda site at the Wanghŭng-sa 王興寺 in Puyŏ. According to these inscriptions, Wanghŭng-sa was built in 577, 11 years before the establishment of Asukadera in 588. Based on this data, both Korean and Japanese academics have raised the possibility that Wanghŭng-sa might have provided inspiration or even a model for Asukadera in Japan. In this paper, I briefly examine reliquaries and roof tiles that have been excavated from the wooden pagoda site within the Wanghŭng-sa site and the arrangement and characteristics of the temple's building sites, including the sites of the main hall and corridors. Next, I discuss the similarities and differences between ancient Buddhist temples in Korea and Japan by comparing Wanghŭng-sa with ancient Japanese temples such as Asukadera.

**Keywords:** Buddhism; śarīra reliquaries; wooden pagoda; model for Asukadera; building sites





## 1. Introduction

Research on the temple sites of the Paekche kingdom began with a survey of the temple site in Kunsu-ri 軍守里寺址 in 1935. Ishida Mosaku from the Tokyo Imperial Museum (present-day Tokyo National Museum) led the excavation of the Kunsu-ri temple site and asserted that Paekche temples exerted a considerable influence on ancient Japanese temples (Ishida 1937, pp. 50–51). This hypothesis was based on the identical layouts of a pagoda and main hall at the Kunsu-ri temple site and those at Shitennoji 四天王寺 in Osaka, and a gilt-bronze standing Bodhisattva sculpture unearthed from the Kunsu-ri temple site that was similar to the Asuka-period gilt-bronze Buddha sculpture at Horyūji 法隆寺 (Ishida 1969, pp. 194–204). According to *Nihon shoki* 日本書紀 (*Chronicles of Japan*), the kingdom of Paekche sent high-ranking officials and monks to Japan and offered śarīra 舍利. It also documents that technicians, including temple craftsmen 寺工, pagoda artisans 鑪盤博士, tile experts 瓦博士, and painters 畫工, were dispatched at the time Asukadera 飛鳥寺 was built. As Ishida noted, the excavation of the Kunsu-ri temple site supported these historical records with additional archaeological materials.

The construction of Asukadera in 588 marked the establishment of Japan's first large-scale Buddhist temple. It was a symbolic event signaling the beginning of the Asuka period, the time when the various systems of thought and technologies accompanying Buddhism were being introduced from the Korean Peninsula. Drawing upon these transmitted thoughts and technologies, Japan developed into an ancient country. In this regard, some researchers describe the temple as emblematic of the cultural enlightenment that took place during the Asuka period, with parallels to the Meiji Restoration of modern Japan (Ōhashi 1997, pp. 6–8). The significance of Asukadera has led Japanese historians to seek to

determine which historic sites from the Paekche kingdom might reveal its origins. They have been putting a great deal of effort into tracing these origins since the early twentieth century, but they have not yet formulated satisfactory answers.

An excavation of Asukadera conducted by the Nara National Research Institute for Cultural Properties (NNRICP) from 1956 through 1957 verified the layout of the temple buildings and discovered reliquaries and convex roof-end tiles. It reconfirmed the similarities between Asukadera and Paekche Buddhist temples (NNRICP 1958, pp. 15–48), in that the artifacts and relics unearthed there resemble those from Paekche Buddhist temples. However, they are not perfectly identical. Thus, there have been considerable controversies in Korean and Japanese academia over how to evaluate and interpret the subtle differences between Asukadera and Paekche Buddhist temples (McCallum 2009, pp. 32–40).

More than 25 temple sites from the Paekche kingdom era, including Chŏngnim-sa site 定林寺址 and the temple site in Nŭngsan-ri 陵山里寺址, remain around Puyŏ, the former capital of the Paekche kingdom. The temple site in Nŭngsan-rI was excavated in 1993, the Wanghŭng-sa site 王興寺址 in 2007, and the Mirŭk-sa site 彌勒寺址 in Iksan in 2009. These excavations revealed the layouts of major buildings, including main halls and wooden pagodas, as well as a vast body of important artifacts such as reliquaries (NMK 2020, pp. 126–77). These findings furthered research into Paekche Buddhist temples and unearthed artifacts that provided new perspectives. In particular, reliquaries with inscriptions found at the wooden pagoda site at the Wanghŭng-sa site revealed that Wanghŭng-sa was built in 577, 11 years before the establishment of Asukadera in 588. Based on these inscriptions, there is a possibility that Wanghŭng-sa, founded by King Widŏk 威德王 (r. 554–598), might have provided inspiration or even a model for Asukadera in Japan.

This paper first introduces a research thread on ancient Buddhist temples in Korean and Japanese academia that has been pursued since the excavation of the wooden pagoda site within the precincts of Wanghŭng-sa site in Puyŏ. In the second section of the paper, I intend to examine the findings on major building sites and reliquaries that have been excavated from the Wanghŭng-sa site. The third section will discuss the similarities and differences between ancient Buddhist temples in Korea and Japan by comparing relics and artifacts found at Paekche temple sites, including Wanghŭng-sa, with ancient Japanese Buddhist temples, such as Asukadera. By doing so, this paper argues that the engineers of Paekje Buddhist temples exemplified by Wanghŭng-sa influenced the construction of Asukadera as a project team in terms of building sites, the layout of buildings, śarīra reliquaries, and roof tiles. Furthermore, I hope to elucidate the characteristics of Paekche temples and their significance within the context of the study of ancient temples in East Asia.

## 2. Findings from Excavations of the Wanghŭng-sa Site in Puyŏ

The Wanghŭng-sa site sits opposite Sabi Fortress, with the Paengma-kang River running between. The river flows from north to south along the west side of the fortress. The discovery of Koryŏ-era concave roof tiles with inscriptions reading "Wanghŭng" validated the presumption that the site was indeed Wanghŭng-sa from the Paekche kingdom. According to *Samguk sagi* 三國史記 (*History of the Three Kingdoms*) and *Samguk yusa* 三國遺事 (*Memorabilia of the Three Kingdoms*), Wanghŭng-sa was founded in 600 and completed in 634. These records also document that this temple was situated by a river, its coloring and decorations were splendid and majestic, and a king visited the temple regularly in a boat to perform *haenghyang* 行香 (a ceremony in which monks go around the temple precinct carrying incense burners as part of a Buddhist assembly) (Kim 1996, vol. 27, pp. 256–57 (Mu 2), vol. 27, p. 259 (Mu 35); Iryŏn 2003, vol. 3, p. 77). The Puyŏ National Research Institute of Cultural Heritage (PNRICH) conducted a 15-year excavation from 2000 through 2015 that identified building sites and relics at the site, as seen in Figure 1 (PNRICH 2009, 2016).

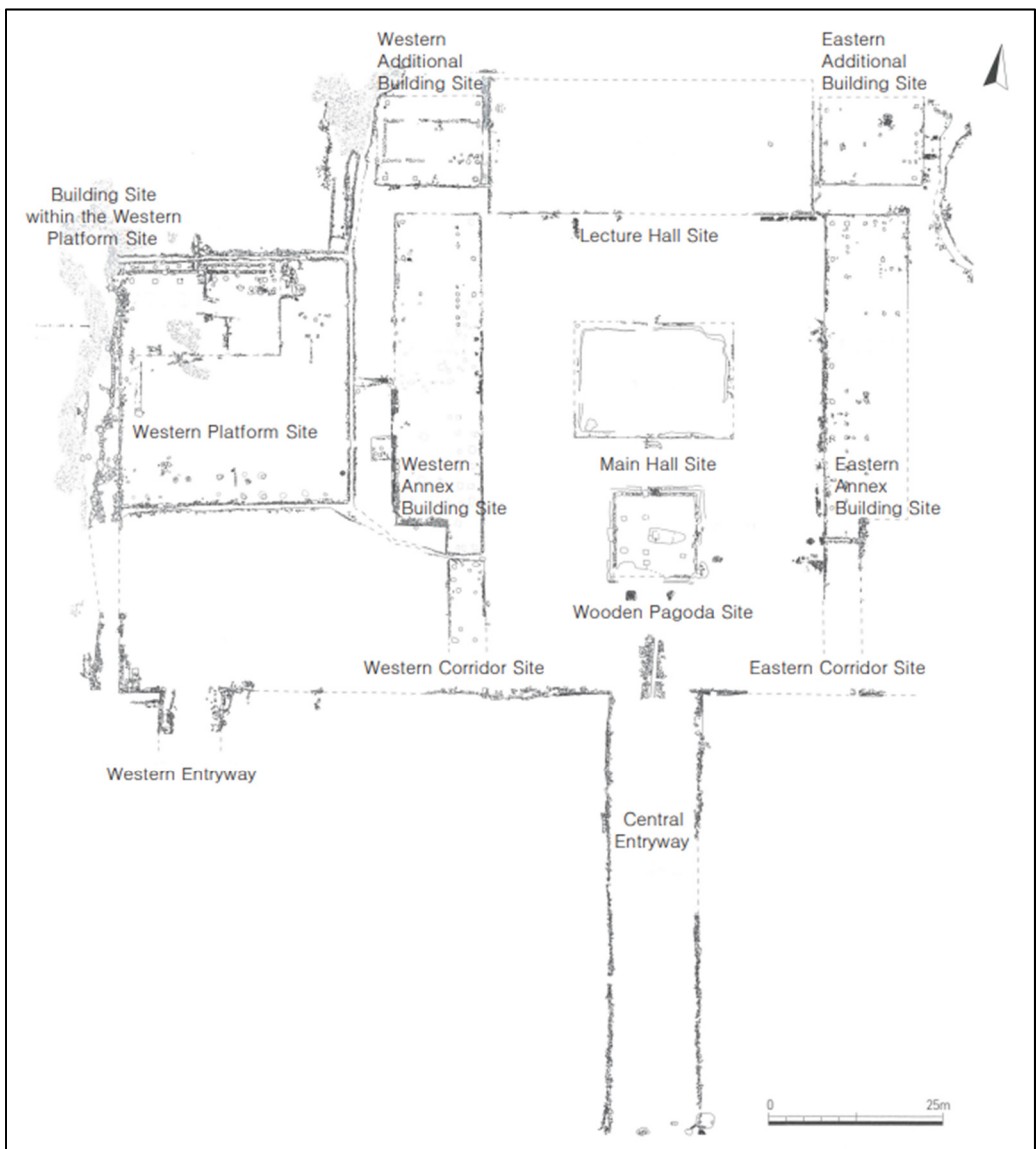

**Figure 1.** The layout of buildings at the Wanghŭng-sa site in Puyŏ. Source: (PNRICH 2017, p. 43).

Since Wanghŭng-sa was originally built on a steep slope descending from north to south, a grand-scale embankment had to be constructed. Its maximum thickness reaches 4.8 m. In the southernmost area of the Wanghŭng-sa site is a central entryway leading to the central precinct and eastern and western embankments. In the northern area, the wooden pagoda, main hall, and lecture hall are aligned with one another. The wooden pagoda was a square structure with sides of 12.2 m, and the main hall behind the wooden pagoda was a rectangular structure running about 22.7 m east–west and 16.6 m north–south. The lecture hall at the rear of the main hall was also a rectangular structure, in this case measuring about 46.9 m east–west and 19.3 m north–south. On both sides of the lecture hall, the sites of two additional buildings, one to each side, were identified.

To the north of the eastern and western corridor sites are two annex building sites, each forming a rectangle spanning about 44.5 m north–south and 12.5 m east–west. These two annex buildings appear to have been set symmetrically and consisted of eight small rooms. A large *ch'imi* (decorative ridge-end tile) 123 cm high was found in the vicinity of the two annex building sites. To the west of the western corridor site is the western platform. There used to be a building measuring about 23.37 m east–west and 13.75 m north–south in the northern portion of the western platform. It was constructed over three occasions. On the southern portion of the western platform was a building measuring

about 2.9 m north–south and 31.9 m east–west, and it was equipped with 11 rooms (Yi 2021, pp. 154–56).

The convex roof-end tiles from the foundation period of Wanghŭng-sa can be divided into two groups (Figure 2). Those in Group A have designs of lotus petals with triangular or heart-shaped ends and are connected to stepless convex tiles. Those in Group B have designs of lotus petals, their ends are adorned with circular projections, and they are connected to tiered convex tiles. Sixteen roof tile kiln sites have been discovered in a hilly area about 150 m east of the central precincts of the Wanghŭng-sa site. This suggests that Wanghŭng-sa was supplied with roof tiles produced at these kilns. A quantitative study on roof tiles from the foundation period of Wanghŭng-sa revealed that these 16 kilns could fire 8602 tiles (concave tiles: 5853; convex tiles: 2749) at a time, and the average quantity of tiles fired in each kiln amounted to 537.6 (Choe 2017, p. 103).

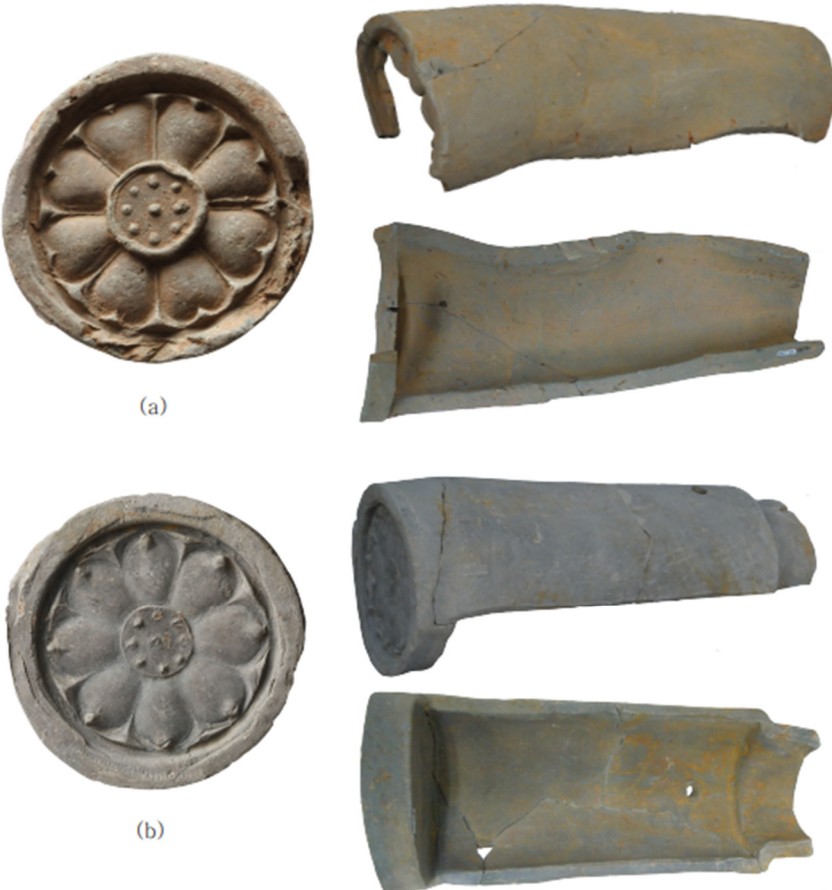

(a)

(b)

**Figure 2.** Roof tiles from the foundation period of the Wanghŭng-sa site ((**a**) Group A, (**b**) Group B)). Source: (PNRICH 2017, p. 140).

A foundation stone 心礎石 for the stylobate of the wooden pagoda was found 50 cm below the soil layer in the central area of the wooden pagoda site (PNRICH 2009, pp. 44–98). The south-central section of this foundation stone has a hole inside, which śarīra reliquaries were enshrined and over which a cover was placed. Inside the hole was a bronze śarīra case with a small silver jar inside it (Figure 3). A smaller gold bottle was found inside the silver jar. However, no śarīra have been discovered. The surface of the bronze śarīra case bears an inscription reading "On the fifteenth day of the second [lunar] month of the *Chŏngyu* year (577), King Ch'ang 昌王 of Paekche founded a temple for the deceased prince, and when he tried to bury two śarīra, they miraculously multiplied into three" (丁酉年二月十五日百濟 王昌爲亡王 子立刹本舍 利二枚葬時 神化爲三).

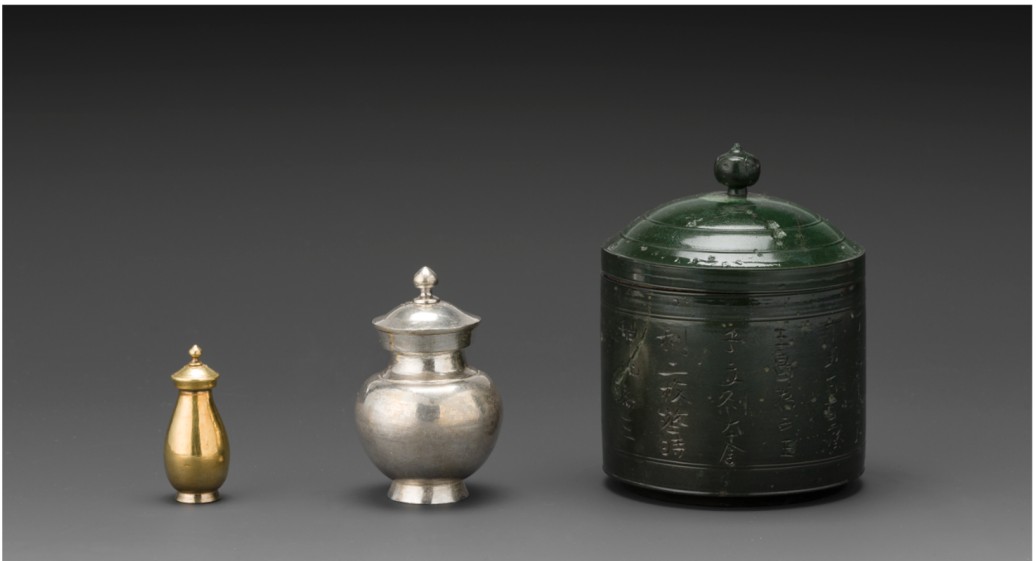

**Figure 3.** Śarīra reliquaries excavated from the wooden pagoda site at the Wanghŭng-sa site. Source: (PNRICH 2017, p. 71).

The 15th day of the second lunar month marks the day when Śākyamuni Buddha entered nirvana. The idea that two śarīra turned into three symbolizes śarīra's miraculous ability to multiply, indicating that the śarīra enshrined under the wooden pagoda of Wanghŭng-sa were the Buddha's true body śarīra (Joo 2018, pp. 64–66). More than 8000 pieces of jewelry and other ornaments (Figure 4) have been excavated from around the foundation stone. They were donated by nobles who participated in the śarīra enshrinement ritual. The materials and compositions are very similar to those of artifacts excavated from the Tomb of King Muryŏng in Kongju and the ancient tombs in Nŭngsan-ri, Puyŏ from the sixth century.

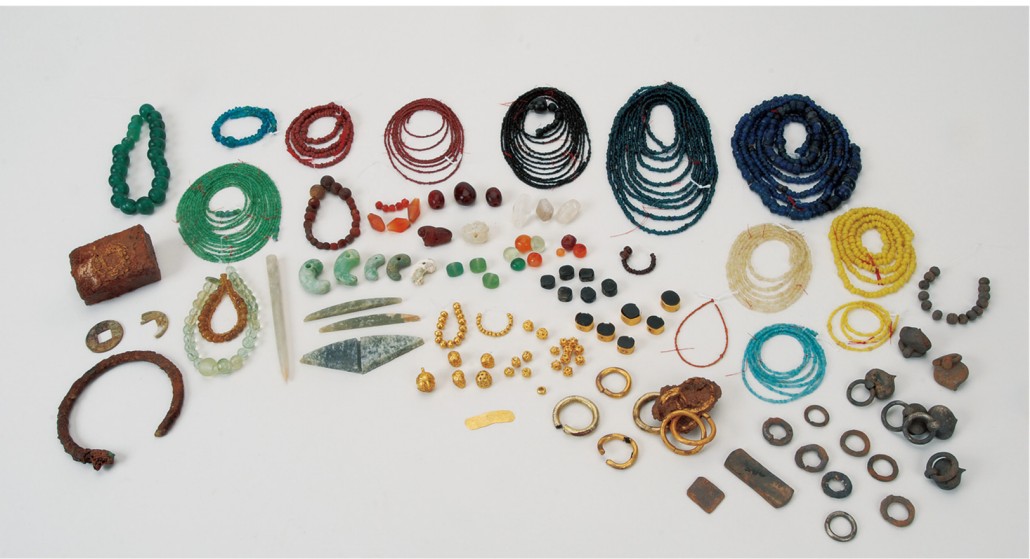

**Figure 4.** Various accessories excavated from the Wanghŭng-sa site. Source: (PNRICH 2017, pp. 82–83).

The inscription on the bronze śarīra case documents that King Widŏk (also known as King Ch'ang), established Wanghŭng-sa in 577. However, *Samguk sagi* records that the construction of the temple began 23 years later in 600 during the reign of King Pŏp 法王 (r. 599–600). There is controversy over how to account for this discrepancy. Some

scholars argue that the two Chinese characters "立利" (*ipch'al*) in the inscription should be interpreted not as referring to the erection of the temple, but to the raising of the central pillar 刹柱 (*ch'alju*) of the wooden pagoda. Based on this interpretation, they suggest that only the wooden pagoda was erected in 577 and the entire temple complex, including the main hall, was completed in 600.

However, according to the investigation of the soil strata in the central area of the temple precinct, the stylobate of the main hall was constructed on the first ground level layer (Figure 5). It also confirmed that the stylobate of the wooden pagoda was built at some point after a second ground level layer applied on top of the main hall's stylobate was dug up. This finding indicates that the construction of the main hall preceded that of the wooden pagoda (PNRICH 2009, pp. 66–67). Thus, the two Chinese characters *ipch'al* in the inscription on the bronze śarīra case from the Wanghŭng-sa site likely do not refer to the establishment of the wooden pagoda, but to the construction of buildings in the central temple precinct.

To resolve this dating discrepancy, some scholars have asserted that Wanghŭng-sa as recorded in *Samguk sagi* actually refers to Mirŭk-sa in Iksan rather than the temple in Puyŏ (Choe 2012, pp. 5–9). According to this theory, the entries for the Wanghŭng-sa and Mirŭk-sa in the historical records are muddled. There is a possibility that the name 'Wanghŭng' could have been used as a common noun rather than being attached to a particular temple. The results of excavations at the Mirŭk-sa site show that the construction period of Mirŭk-sa indeed closely conforms to that of Wanghŭng-sa as recorded in *Samguk sagi.* This argument is somewhat flawed, however, as no navigable waterways are found in the vicinity of the Mirŭk-sa site but an entry in *Samguk sagi* states that the king came to Wanghŭng-sa in a boat. Nevertheless, primary historical materials and unearthed artifacts indicate that Wanghŭng-sa in Puyŏ served as the Paekche kingdom's royal temple and was built in 577.

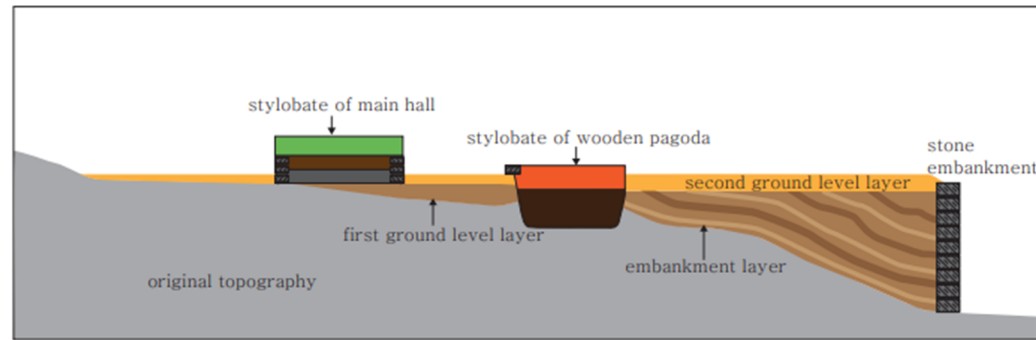

**Figure 5.** Schematic diagram of the construction process of the main hall and wooden pagoda at Wanghŭng-sa. Source: (PNRICH 2017, p. 66).

## 3. A Comparison between Paekche Buddhist Temples and Asukadera

The Paekche kingdom transmitted Buddhism to Japan. The year when Buddhism was introduced to Japan is presumed to be either 538 or 552. Recently, however, some scholars have formulated an alternative hypothesis that Buddhism might have been transmitted from Paekche to Japan for political reasons in the 540s when Paekche and Japan were negotiating sovereignty over the Imna 任那 region. King Sŏng 聖 (r. 523–554) of the Paekche kingdom required military aid and cooperation from Japan in order to achieve its goal of regaining its former territory around the Han-Kang River. It has been believed that King Sŏng engaged in diplomatic activities by means of Buddhism, referring to the Buddhist governing policy instituted by Emperor Wu 武帝 (r. 502–549) of the Liang Dynasty of the Chinese Southern Dynasties (Choe 2011, pp. 81–85).

The year 577 marked a turning point in the early process of Japan's adoption of Buddhism. *Nihon shoki* records as follows: "[In the eleventh lunar month of 577], the king of the Paekche kingdom gave envoys returning [to Japan], including Ōwake no okimi 大別

王, Buddhist images and several treatises on Buddhist scriptures, and [ordered] six people, including a Vinay monk 律師, a Sŏn master 禪師, a Buddhist nun 比丘尼, a sorcerer 呪禁師, a Buddhist image maker 造佛工, and a temple carpenter 造寺工 [to accompany them]. [The images] were enshrined at Ōwake no okimi no dera 大別王寺 in Naniwa 難波 [the present-day Osaka region]" (Taro et al. 1995, vol. 20, p. 140 (Bidatsu 6)). These records have drawn more attention since a śarīra case enshrined on the 15th day of the second lunar month of 577 was unearthed from the wooden pagoda site at the Wanghŭng-sa site in Puyŏ. It is difficult to discover more about King Owake and Owakeoji Temple as there are no other available records. Nevertheless, among the six people who went to Japan in 577, was a Buddhist image maker 造佛工 (*chobulgong*), as well as a temple carpenter 造寺工 (*chosagong*), who can be viewed as an architectural engineer. They seemed to have had some sort of relationship with the technicians involved in the construction of Wanghŭng-sa founded in the second lunar month of 577.

Soga no Umako 蘇我馬子, who won a civil war with the Mononobe clan in 587, initiated the establishment of Asukadera. The construction began in 588 when high-ranking officials, Buddhist monks, and several technicians from the Paekche kingdom were dispatched to Japan. The construction process of Asukadera is elucidated in detail in several historical records, including *Nihon shoki* (Taro et al. 1995, vol. 21, pp. 168–69 (Sushun 1)). The *sagong* 寺工 (temple craftsman) and *sasa* 寺師 (temple master) documented in these historical records refer to technicians related to wooden architecture, such as a *chosagong* 造寺工 (temple carpenter); the *nobanbaksa* 露盤博士 (pagoda artisan) and *nubansa* 鏤盤師 (pagoda master) both indicate a metal artisan who produced the metal finial of the upper section of a Buddhist pagoda; the *wabaksa* 瓦博士 (tile expert) and *wasa* 瓦師 (tile master) refer to the artisans in charge of producing roof tiles; and the *hwagong* 畫工 (painter) refers to an artisan responsible for creating designs and paintings in and outside the temple hall. Technicians in such diverse fields were dispatched from the Paekche kingdom as a single project team (Yi 2015, pp. 250–52).

The surviving relics and artifacts discovered during the excavations indicate Asukadera's relation to Paekche Buddhist temples. The foundations of the wooden pagoda and three main halls at Asukadera were constructed using the rammed earth technique called *P'anch'uk* 版築. A civil engineering method for improving soft ground, the rammed earth technique, involves laying down alternating materials with different properties, including earth and sand, and compressing the layers by pounding with wooden poles. Although known to be used mostly in the construction of ramparts or embankments, this technique was often applied when constructing heavy, large-scale buildings with tiled roofs and foundation stones, including high-rising wooden pagodas (Aoki 2017, pp. 115–23). The application of rammed earth has been verified not only at the wooden pagoda site of Asukadera, but also at Paekche Buddhist temples. This is presumed to be the influence of temple carpenters and temple masters from the Paekche kingdom.

Similarities between Asukadera and Paekche Buddhist Temples can also be found in the stylobate construction method applied to buildings with tiled roofs. The central main hall of Asukadera was built on a stylobate constructed with polished stone slabs in a post-and-lintel structure, while the stylobates of the eastern and western main halls were made of naturally found stones piled up in a random fashion. Interestingly, small foundation stones were placed above the lower portions of the stylobates for the eastern and western main halls. Among the known Paekche Buddhist temple sites, the use of a post-and-lintel structure in the stylobates can be found at the sites of the main hall and wooden pagoda within the precincts of the Nŭngsan-ii site. Double stylobates with their lower portions covered by small foundation stones are found at the main hall sites within the precincts of the Chŏngnim-sa site (PNRICH 2011, pp. 109–15) and Kunsu-ri temple site (PNRICH 2010, pp. 44–61), and at the building sites on Kŭmsŏng Mountain Building Site (PNM 1992, pp. 14–19).

Double stylobates with their lower portions covered by small foundation stones have been also found at the wooden pagoda site (late 5th–early 6th century) within the

precincts of the Ch'ŏngam-ri temple from the Koguryŏ kingdom (NNRICP 1958, pp. 43–44). Examples of the use of double stylobates and small foundation stones have been found earlier in Koguryŏ than in Paekche. However, based on the pertinent historical records, many researchers support the theory that the architectural technique used to build the eastern and western main halls of Asukadera was transmitted not through Koguryŏ, but through Paekche. This point should be taken into consideration when tracing the origins of the three main halls of Asukadera, which will be discussed later in this paper (McCallum 2009, pp. 48–53).

The shape and installation method of the foundation stone in the wooden pagoda site of Asukadera as well as the reliquaries discovered from inside the foundation stone reveal the influence of the Paekche kingdom. A wide variety of foundation stones installed in diverse manners to support the central pillars of wooden pagodas have been discovered at ancient Buddhist temple sites in Japan. The installation method for such foundation stones evolved from burying them deeply below the ground level to half-burying them in the ground, and later they were set at ground level (Sagawa 2010, pp. 168–70). Notably, the foundation stone for the wooden pagoda at Asukadera was found buried below the ground level, as also discovered at the wooden pagoda sites at the Nŭngsan-ri, Kunsu-ri, and Wanghŭng-sa sites (McCallum 2009, pp. 58–61).

Traces of an intentionally oblique ramp for erecting a central pillar have been found at the wooden pagoda sites within the precincts of the Kunsu-ri and Wanghŭng-sa sites, and similar types of ramps were found at the wooden pagoda sites of Asukadera, the Wakakusa site 若草伽藍, and Chūgūji 中宮寺 (Ōwaki 2016, pp. 316–24). Moreover, a similar enshrinement method for reliquaries in the foundation stones of a wooden pagoda can be seen, not just at Asukadera, but at the Wanghŭng-sa and Kua-ri temple sites (PNM 2016, pp. 249–51) as well (Figure 6). The practice of creating a hole in a foundation stone and placing śarīra inside is thought to have been first implemented in the Paekche kingdom; the placement of a cover over the square hole for śarīra can be found both in the Paekche kingdom and Japan (Ōwaki 2013, pp. 264–73).

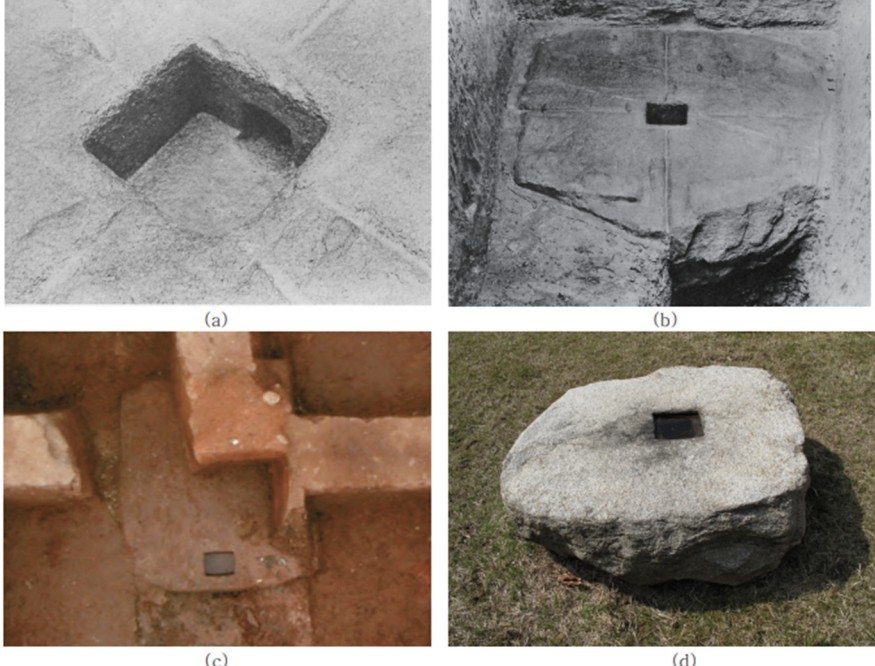

**Figure 6.** Foundation stones of Buddhist pagodas from the Paekche kingdom and Japan ((**a**,**b**) Asukadera, (**c**) Wanghŭng-sa site; (**d**) Kua-ri temple site). Source: (**a**,**b**) (NNRICP 1958, p. 132), (**c**) (PNM 2016, p. 251), (**d**) (PNRICH 2017, pp. 73–74).

As can be seen in Figure 7a, similar types of glass beads, jade, and gold and silver items have been unearthed from the wooden pagoda sites at the Wanghŭng-sa site and at Asukadera. In the case of the glass beads, even their chemical compositions are identical (AHM 2013, p. 32). However, among the artifacts found at the wooden pagoda site at Asukadera are objects reminiscent of tomb furnishings, such as a small ornamental knife, armor (Figure 7b) made from small iron plates, and a horse harness, none of which are included in reliquaries in the Paekche kingdom. When these tomb furnishings were first discovered, a close relation to ancient burial mounds 古墳 (*kofun* in Japanese) was highlighted to the degree that the foundation stone was thought to be a stone chamber with a walk-in horizontal entrance from the Kofun period (ca. 300–538). This discovery attests to the originality of ancient Japanese Buddhist temples (Tsuboi 1985, pp. 32–38). Even so, as reliquaries in the Paekche kingdom are very similar to artifacts excavated from tombs, they should be perceived as offerings donated by nobles who participated in śarīra enshrinement rituals (Joo 2018, pp. 57–60). Therefore, the inclusion of objects similar to tomb furnishings inside the wooden pagoda of Asukadera can be also understood as a result of the Paekche influence (Joo 2018, pp. 57–60).

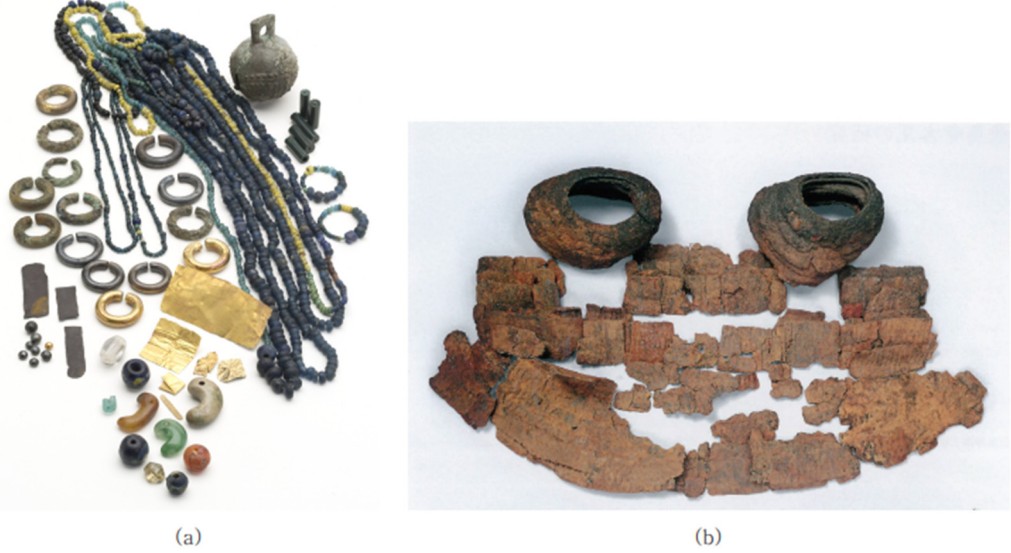

(a)             (b)

**Figure 7.** Various accessories (**a**) and armor (**b**) excavated from the wooden pagoda site at Asukadera in Japan. Source: (AHM 2013, pp. 2, 5).

The roof tiles from the foundation period at Asukadera are also a result of the involvement of Paekche tile experts in the construction of the temple. These roof tiles can largely be classified into two types according to the design of the lotus petals: (A) the *hanagumi* 花組 type with heart-shaped lotus petals, and (B) the *hoshigumi* 星組 type with dots around the edges of the lotus petals (Figure 8). These two types of roof tiles are connected to different convex tiles and used in different buildings (Hanatani 2000, pp. 28–30). The roof tiles from the foundation period of Wanghŭng-sa bear similarities to the roof tiles from the foundation period of Asukadera in terms of the designs and connected convex tiles (Sagawa 2010, p. 180; Yi 2015, pp. 237–38). They were produced at a kiln designed to fire roof tiles in the eastern area of the temple precinct. This roof tile production system was established under the influence of the Paekche kingdom (McCallum 2009, pp. 64–69).

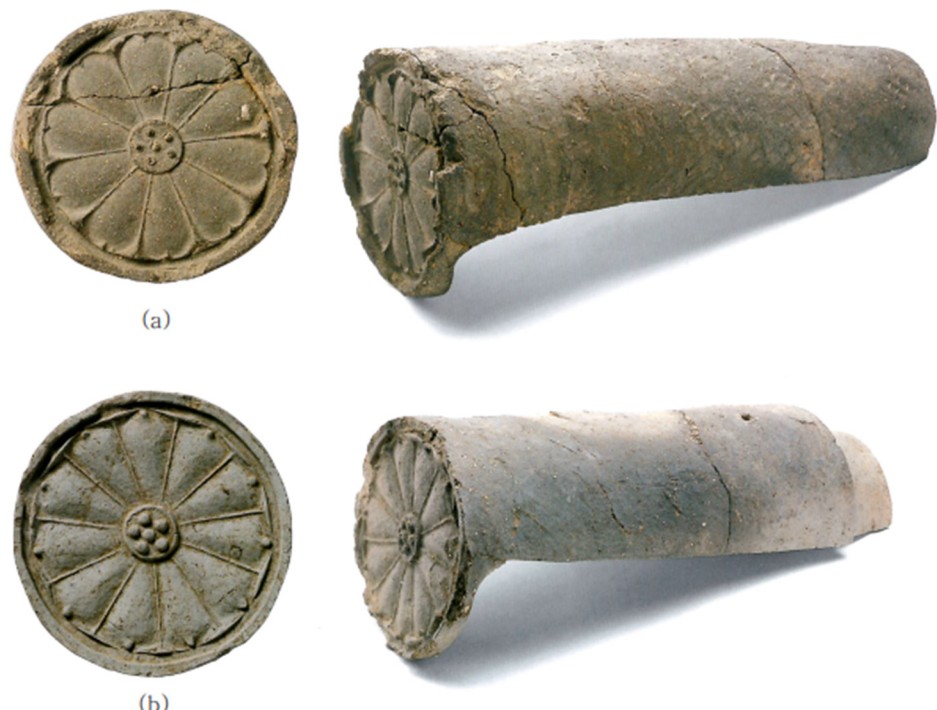

**Figure 8.** Roof tiles from the foundation period of Asukadera in Japan ((**a**) *Hanaumi*, (**b**) *Hoshigumi*). Source: (AHM 2013, p. 7).

Despite their many similarities, Asukadera and Paekche Buddhist temples differ in the layout of the buildings. The three main halls are placed around a wooden pagoda at Asukadera, one to the east, another to the west, and the other on the north (Figure 9). On the Korean Peninsula, this layout with three main halls can be observed only at temples sites from the Koguryŏ kingdom, including the Ch'ŏngam-ri temple site (Koizumi 1940, pp. 5–19). Accordingly, the layout of Asukadera had been suggested as having been influenced by the Koguryŏ kingdom (NNRICP 1958, p. 39; Tsuboi 1985, pp. 46–48).

However, beyond its layout, other elements related to Koguryŏ Buddhist temples are difficult to find. Moreover, the Asukadera pagoda is square-shaped rather than taking the octagonal form commonly found in Koguryŏ pagodas. Several theories have been proposed to account for this. Some presume that the three halls might have been constructed over two different time periods, while others suggest that Koguryŏ ideas might have passed through Paekche and then been introduced to Japan (Kim 1990, pp. 127–38). There is also an expectation that a similar layout with three main halls might be discovered in the regions ruled by the Paekche kingdom. As two annex building sites have been recently found in the north of the eastern and western corridor sites within the precincts of the Wanghŭng-sa site, the Japanese scholar Sagawa Masatoshi asserts that these two annex buildings were converted into the eastern and western main halls at Asukadera in Japan (Sagawa 2010, pp. 163–64).

Sagawa's assertion about the relation of Asukadera's eastern and western main halls to annex buildings in the north of the eastern and western corridors within the precincts of Paekche Buddhist temples, including Wanghŭng-sa, aroused a sensation within academia. However, given that the annex building sites commonly found at Paekche Buddhist temples consist of several rooms and are vertically elongated while deviating from the central axis, these annex buildings should be viewed as a form of *Sŭngbang* 僧房 or monks' quarters that also functioned as a public space (Yi 2014, pp. 98–100). This hypothesis can be proven by the excavation of wooden strips from the south of the middle gate site within the Nŭngsan-ri temple site (PNM 2007, pp. 238–59), the analysis of the inscriptions written on the wooden strips, and the monks' quarters in the Mirŭk-sa site (Yi 2014, pp. 213–14).

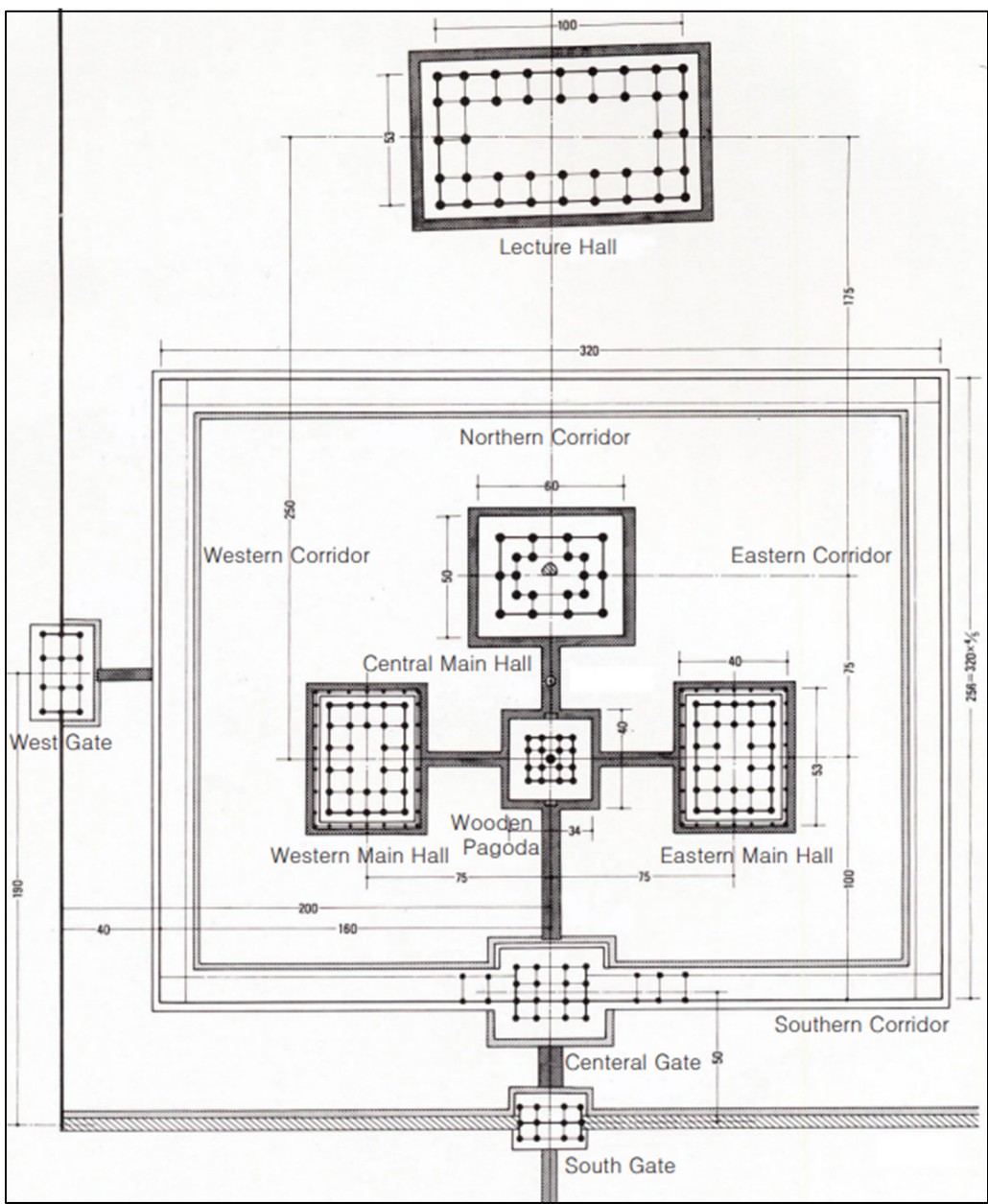

**Figure 9.** The layout of buildings within Asukadera in Japan. Source: (AHM 2013, p. 18).

Regarding the difference in the layouts of Asukadera and Paekche Buddhist temples, I believe that we need to pay more attention to a building site to the east of the eastern corridor site at the Kunsu-ri temple site and a building site on the western platform to the west of the western corridor site at the Wanghŭng-sa site. These sites can be viewed as evidence of new elements, including multi-compound layouts, incorporated into late sixth-century Paekche Buddhist temples. The Kunsu-ri temple site and Wanghŭng-sa site are also equipped with east and west precincts, as seen in the Mirŭk-sa Site complex (Yi 2015, pp. 178–82). Around the time of the transfer of the Paekche kingdom's capital to Sabi in 538, the material culture of the Koguryŏ kingdom was exerting a major impact on the Puyŏ region, as demonstrated by the discovery of Koguryŏ-style earthenware, roof tiles, metal craftworks, *ondol* (traditional underfloor heating system), and mural paintings depicting the guardian deities of the four directions.

Considering the Koguryŏ-influenced elements identified in the Paekche kingdom, it would be more reasonable to presume that some aspects of Koguryŏ culture might have been internalized in the Paekche kingdom and then introduced to Japan than to

believe that only the three main halls of Asukadera were directly impacted by the Koguryŏ kingdom, even if three main halls similar to those of Asukadera have not been found in the regions ruled by the Paekche kingdom (Yi 2015, pp. 169–71). In contrast to the construction of ancient tombs, ancient Buddhist temples relied upon mutual technical cooperation among countries. At the time when Asukadera was built as Japan's first large-scale Buddhist temple, people in Japan could not have selectively imported temple construction technologies from different foreign countries. In this light, the construction of Asukadera should be understood as a project completed by a team of technicians from the Paekche kingdom.

Unlike at Asukadera, annex building sites have been found in Toyoradera 豊浦寺 for fully ordained Buddhist nuns. Along with Asukadera, Toyoradera has been regarded as one of the oldest Japanese Buddhist temples. The re-examination of the preexisting excavation results revealed vestiges of annex buildings in the east of the lecture hall in the style of Shitennoji in Osaka (Yi 2015, pp. 205–7). At Shindohaiji 新堂廃寺 in Kawachi 河内, eastern and western buildings were once positioned to the north of the eastern and western corridors. Such building sites are classic examples of annex building sites as found at Paekche Buddhist temples (Yi 2015, pp. 208–9). Taking all this into account, it can be suggested that Paekche Buddhist temples such as Wanghŭng-sa continued to exert a considerable impact on ancient Japanese Buddhist temples, including not only Asukadera, but temples established afterwards.

## 4. Conclusions

In the excavation report on Asukadera published in 1958, Japanese academics noted that relics and artifacts excavated from Asukadera are highly similar to those found at Paekche Buddhist temples, but that Asukadera's three main halls show the influence of the Koguryŏ kingdom (NNRICP 1958, p. 39). This view is one of several theories put forward based on the fact that at the time, the Ch'ŏngamri Temple Site from the Koguryŏ kingdom was the only known example with three main halls (Koizumi 1940, pp. 5–19). Some scholars who embraced this view as an established theory affirmed that Asukadera reflects international cultural elements from the Paekche and Koguryŏ kingdoms and that people in Japan at the time had the capability to adopt or reject different cultural elements. However, it is difficult to imagine that people in Japan at the time would have the freedom to make such choices when building a temple, which required great expense and a long time to complete, by inviting technicians from overseas. More specifically, the supposition that at the time when the addition of two more main halls was decided, people in Japan could afford to spontaneously obtain the necessary wooden and stone materials and roof tiles is unfounded.

In contrast, some Korean researchers have maintained that the dissemination of Paekche culture to Asukadera proves the superiority of ancient Korean culture and that Korea introduced culture to Japan unilaterally as some sort of favor. However, cultural exchanges between regions must have been mutually beneficial, as they are now. One-sided aid from a particular country to another is difficult to imagine.

Why would the Paekche kingdom dispatch technicians to Japan and support the establishment of temples at the time? In the mid- and late-sixth century, the Paekche kingdom transmitted Buddhism and temple construction technology to Japan as a means to strengthen its diplomatic hegemony over the area under changing international circumstances marked by the development of the Silla kingdom and the establishment of the Sui Dynasty 隋 (581–618) in China (Yi 2015, pp. 212–13).

Among the known Paekche Buddhist temples, Wanghŭng-sa, founded in 577, is chronologically the nearest to Asukadera, which was completed in 588. Similarities between these two temples should not be surprising. However, as discussed above, the understanding and interpretations of materials regarding these two temples remain biased in favor of either Korea or Japan, and failed to disconnect themselves from nationalism. Although materials related to Buddhist temples in the Southern Dynasties 南朝 (420–589) in

China, which are known to have had a substantial influence on Paekche Buddhist temples, are extremely rare, the possibility that a temple with three main halls will be identified someday cannot be eliminated. Moreover, the discovery of new materials is expected.

**Funding:** This research received no external funding.

**Data Availability Statement:** Not applicable.

**Conflicts of Interest:** The author declares no conflict of interest.

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
