# Peer review of "Tracing the Origin of Japan’s First Buddhist Temple: Japan’s Asukadera Viewed through the Lens of the Korean Paekche Kingdom Temple Site of Wanghŭng-sa"

_religions, doi:10.3390/rel13090846_

Round 1
Reviewer 1 Report
“Tracing the Origins of Japan’s First Buddhist Temple: Japan’s Asakadera Viewed Though the Lens of the Korean Paekche Kingdom Temple Site of Wanghŭngsa”
This essay shows great promise. It succinctly treats important connections between these two early Buddhist temples. There are only a few issues that the author needs to address in this relatively short paper. First, the author seems completely unaware of a detailed study of Asukadera published in English. Although this study is based on much of the Japanese scholarship known to the author, the author needs to be familiar with this English-language study and cite it in his essay:
McCallum, Donald F. The Four Great Temples: Buddhist Archaeology, Architecture, and Icons of Seventh-Century Japan. Honolulu: University of Hawai‘i Press, 2009. [Chapter 1 of this book is a detailed study of Asukadera, pp. 23–82]
The final paragraph of the introduction (lines 69-78) explains what the author will do in the article. The author should also include some general statements of the results of his study. In other words, he should briefly state in what ways Asukadera shows influence from Wanghŭngsa. This should be a brief statement of the author’s key findings as described in the conclusion (lines 393-429).
The other major issues that needs to be addressed are stylistic issues. The word “temple” is far overused in this paper. The Sino-Korean phoneme sa, the Sino-Japanese phoneme ji, and the Japanese phoneme dera all mean “temple.” Therefore, it is repetitive and redundant to repeat the word temple after these phonemes. The context of the paper makes it clear that the author is talking about Buddhist temple sites; therefore, continually using the word “temple” becomes laborious to a native English speaker’s ears and detracts from the strengths of the essay. In the attached pdf, this reviewer has crossed out words that do not need to be in the article following the above reasoning.
This paper uses the McCune-Reischauer Romanization system for Romanizing Korean. This system uses the diacritical mark known as the breve. In many places in the paper, breves are problematically placed in spots next to the letter over which it should be placed (e.g. see Wanghŭngsa on lines 10, 12, 14, 16, and more) or not fully over the word (e.g., see Puyŏ, on line 58, 71; Widŏk on line 156, etc.). In the attached pdf, this reviewer has circled the cases where the breve needs to be fixed.
Some terms in the paper need to be corrected to the McCune-Reischauer system or are Japanese words that need to be broken following the standards of Western academe, or are Sanskrit names that should follow proper academic conventions:
p. 2, line 82: Goryeo-era should be Koryŏ-era
p. 5, line 144: Shakyamuni should be Śākyamuni
p. 6, line 195: Imna does not need to be italicized—it is a proper noun.
p. 6, line 197: Hangang River should be Han River (because -gang means river); and in Western languages the river that runs through Seoul Korea is merely the Han River.
p. 7, line 203: Owakenookimi should be Owake no okimi
p. 7, line 204: Buddhist monk 律師 should be “vinaya master” or “precepts master”
p. 7, line 213: a pulsa 佛師 usually means “Buddhist master”; the author should explain better why this refers to a master of images in this case
p. 7, line 236: panchuk should be p’anch’uk
p. 8, line 257-258: Kŭmsŏngsan Mountain should Kŭmsŏng Mountain
p. 9, line 315: both hanagumi and hoshigumi should be italicized.
p. 11, line 353: seungbang should be sŭngbang in McCune-Reischauer
p. 11, line 356: Nŭngsanri should be either Nŭngsan-ri or Nŭngsalli in McCune-Reischauer
p. 12, line 398: Ch’ŏngamri should be either Ch’ŏngam-ri or Ch’ŏngamni in McCune-Reischauer
p. 14, line 473: Wangheungsa should be Wanghŭngsa
p. 14, line 516: Wangheungsa should be Ōkōji (Japanese pronunciation since essay in Japanese)
The notes, p. 13, lines 436-443, should follow a standard citation style.
1. Samguk sagi 27:page? (Mu 35); Samguk yusa 3:page.
2. Nihon shoki 20:page? (Bidatsu 6).
3. Nihon shoki 21:page? (Sushun 1).

Author Response
Response to Reviewer 1 Comments
Point 1 : This essay shows great promise. It succinctly treats important connections between these two early Buddhist temples. There are only a few issues that the author needs to address in this relatively short paper. First, the author seems completely unaware of a detailed study of Asukadera published in English. Although this study is based on much of the Japanese scholarship known to the author, the author needs to be familiar with this English-language study and cite it in his essay:
McCallum, Donald F. The Four Great Temples: Buddhist Archaeology, Architecture, and Icons of Seventh-Century Japan. Honolulu: University of Hawai‘i Press, 2009. [Chapter 1 of this book is a detailed study of Asukadera, pp. 23–82]
Response 1 : I really appreciate your detailed comments and advice. I have tried to supplement what you have pointed out by citing the book you introduced. I have cited this book to discuss scholarship on Asukadera (particularly regarding double stylobates of eastern and western main halls, roof tiles from the foundation period, and wooden pagoda site excavated relics).
Point 2 : The final paragraph of the introduction (lines 69-78) explains what the author will do in the article. The author should also include some general statements of the results of his study. In other words, he should briefly state in what ways Asukadera shows influence from Wanghŭngsa. This should be a brief statement of the author’s key findings as described in the conclusion (lines 393-429).
Response 2 : Thank you for your comment. In order to clarify key findings of this paper, the following sentence has been added to the text: “By doing so, this paper argues that the engineers of Paekje Buddhist temples exemplified by Wanghŭng-sa influenced the construction of Asukadera as a project team in terms of building sites, the layout of buildings, śarīra reliquaries, and roof tiles.”
Point 3 : The other major issues that needs to be addressed are stylistic issues. The word “temple” is far overused in this paper. The Sino-Korean phoneme sa, the Sino-Japanese phoneme ji, and the Japanese phoneme dera all mean “temple.” Therefore, it is repetitive and redundant to repeat the word temple after these phonemes. The context of the paper makes it clear that the author is talking about Buddhist temple sites; therefore, continually using the word “temple” becomes laborious to a native English speaker’s ears and detracts from the strengths of the essay. In the attached pdf, this reviewer has crossed out words that do not need to be in the article following the above reasoning.
Response 3 : I have deleted the “temple” to avoid redundancy.
Point 4 : This paper uses the McCune-Reischauer Romanization system for Romanizing Korean. This system uses the diacritical mark known as the breve. In many places in the paper, breves are problematically placed in spots next to the letter over which it should be placed (e.g. see Wanghŭngsa on lines 10, 12, 14, 16, and more) or not fully over the word (e.g., see Puyŏ, on line 58, 71; Widŏk on line 156, etc.). In the attached pdf, this reviewer has circled the cases where the breve needs to be fixed.
Response 4 : The text has been modified in the proper McCune-Reischauer Romanization format.
Point 5 : Some terms in the paper need to be corrected to the McCune-Reischauer system or are Japanese words that need to be broken following the standards of Western academe, or are Sanskrit names that should follow proper academic conventions:
Response 5 : I have checked the Japanese and Sanskrit words and corrected mistakes.
Point 6 : p. 7, line 213: a pulsa 佛師 usually means “Buddhist master”; the author should explain better why this refers to a master of images in this case.
Response 6 : I have deleted pulsa 佛師 to avoid confusion. For the same reason, 寺師 (sasa) and 寺工 (sagong) have been deleted in some parts of the paper.
Point 7 : The notes, p. 13, lines 436-443, should follow a standard citation style.
Response 7 : Note and references have been modified to the standard citation style.
Reviewer 2 Report
This article is exceptionally well written and argued. Principally it compares two archaeological sites, and draws good conclusions from the evidence provided. It has excellent and helpful illustrations. This article can be published as is without any alteration of content or writing.
However, there are a number of editorial matters which must be resolved before it is presented for publication. The issues include the following:
1) Words which are generic should be in lower case, i.e., xxx Temple Site is xxx temple site, or xxx Temple is xxx temple. 'Sa' is temple so adding that with a capital is redundant.
2) It is preferable that in Romanisation class words (sa for temple, kang for river, etc) should be attached by a hyphen to the name. For example, Wanghŭngsa should be Wanghŭng-sa, and so on.
3) In the text and in the bibliography, terms and words which are Japanese should be Romanised according to Japanese pronunciation and not Korean.
4) Some specialised terms, such as sari, should be explained at the first mention.
5) On line 82, Goryeo-era should be Koryŏ-era.
6) In the bibliography, when an author's name is given, it should be given in the standard M-R form and then in brackets the author's spelling given if different followed by Chinese characters if known. Thus Lee, Byongho should be Yi, Pyŏngho (Lee, Byongho, XXX). The form (Lee 2014) should be removed from the bibliographical entry
7) In the bibliography, Sagawa's book should have Wangheungsa written according to the Japanese pronunciation of the Chinese characters.
8) In many cases in my text, the micron (short vowel marker) is not over the 'o' or 'u' but is floating to the side. This needs to be corrected.
9) The article must be thoroughly proofread to ensure consistency throughout. This is an excellent article and it should be in the best form to make the impact it deserves.
Author Response
Response to Reviewer 2 Comments
Point 1: Words which are generic should be in lower case, i.e., xxx Temple Site is xxx temple site, or xxx Temple is xxx temple. 'Sa' is temple so adding that with a capital is redundant.
Response 1: Thank you very much for the reviewer's constructive comments and advice. I have removed the repeatedly used “temple” and modified duplicate capital letters.
Point 2: It is preferable that in Romanisation class words (sa for temple, kang for river, etc) should be attached by a hyphen to the name. For example, Wanghŭngsa should be Wanghŭng-sa, and so on.
Response 2: I have added a hyphen according to the proper Romanization.
Point 3: In the text and in the bibliography, terms and words which are Japanese should be Romanised according to Japanese pronunciation and not Korean.
Response 3: I have modified the Japanese notation in the text according to Japanese pronunciation.
Point 4: Some specialised terms, such as sari, should be explained at the first mention.
Response 4: I have added some explanations for such specialized terms.
Point 5: On line 82, Goryeo-era should be Koryŏ-era.
Response 5: I have modified it.
Point 6: In the bibliography, when an author's name is given, it should be given in the standard M-R form and then in brackets the author's spelling given if different followed by Chinese characters if known. Thus Lee, Byongho should be Yi, Pyŏngho (Lee, Byongho, XXX). The form (Lee 2014) should be removed from the bibliographical entry.
Response 6: The references have been modified following the appropriate format.
Point 7: In the bibliography, Sagawa's book should have Wangheungsa written according to the Japanese pronunciation of the Chinese characters.
Response 7: I have modified it to Ōkoji, which is the Japanese pronunciation of Wanghŭng-sa.
Point 8: In many cases in my text, the micron (short vowel marker) is not over the 'o' or 'u' but is floating to the side. This needs to be corrected.
Response 8: All missing the microns (short vowel marker) have been added and corrected.